# Chemical Profiling, Antioxidant and Antimicrobial Activities, and In Silico Evaluation of *Gardenia jasminoides* Essential Oil

**DOI:** 10.3390/plants14071055

**Published:** 2025-03-28

**Authors:** Mohammed Kara, Nouha Haoudi, Nor El houda Tahiri, Fatima Zahra Rhebbar, Reda El Mernissi, Amine Assouguem, Hamid Slali, Jamila Bahhou

**Affiliations:** 1Laboratory of Biotechnology, Conservation and Valorisation of Bioresources (LBCVB), Faculty of Sciences Dhar El Mehraz, Sidi Mohamed Ben Abdallah University, P.O. Box 1796 Atlas, Fez 30000, Morocco; haoudinouha302@gmail.com (N.H.); jamila.bahhou@usmba.ac.ma (J.B.); 2Fruit Trees and Vine Research Program, National Institute for Agricultural Research (INRA), P.O. Box 578, Meknes 50001, Morocco; 3Natural Resources and Sustainable Development Laboratory, Department of Biology, Faculty of Sciences, Ibn Tofail University, P.O. Box 133, Kenitra 14000, Morocco; norelhouda.tahiri@uit.ac.ma (N.E.h.T.); fatimazahra.rhebbar@uit.ac.ma (F.Z.R.); 4Molecular Chemistry and Natural Substances Laboratory, Faculty of Science, University Moulay Ismail, Meknes 50050, Morocco; re.elmernissi@edu.umi.ac.ma; 5Ethnopharmacology and Pharmacognosy Team, Faculty of Sciences and Technology, Errachidia, University Moulay Ismaïl, Meknes 50050, Morocco; assougam@gmail.com; 6Laboratory of Scientific Innovation in Sustainability, Environment, Education and Health in Era of AI (ISDEESIA), Department of Biology-Geology, Ecole Normale Supérieure (ENS), Sidi Mohamed Ben Abdellah University (USMBA), Fes 30000, Morocco; hamidslali@gmail.com

**Keywords:** medicinal plants, *Gardenia jasminoides*, secondary metabolites, phytochemistry, essential oil

## Abstract

Aromatic and medicinal plants have been integral to human civilization for thousands of years, serving not only as vital components in traditional and modern medicine but also as sources of captivating fragrances that enhance our sensory experiences. The main objective of this study was to explore the chemical composition, antioxidant and antimicrobial properties, and in silico molecular docking attributes of *Gardenia jasminoides* essential oil (*GJEO*). The chemical compositions were determined using gas chromatography–mass spectrometry (GC-MS) analysis. The antioxidant activity was determined by 2,2-Diphenyl-1-picrylhydrazyl (DPPH) and total antioxidant capacity (TAC) test. The antimicrobial activity was tested in vitro using three microbial strains (*Escherichia coli*, *Pseudomonas aeruginosa*, and *Staphylococcus aureus*), and two fungal strains (*Candida albicans* and *Aspergillus niger*). In silico analysis by molecular docking was used to determine the interaction types of topoisomerase II receptors and the most important antioxidant and antimicrobial compounds (Eugenol, Methyleugenol, and α-Terpineol ligands). The obtained results highlight the presence of 25 volatile compounds including 5 new detected compounds: Methyleugenol (15.41%), 1-Undecyne (3.4%), 2,6,10-Dodecatrien-1-ol, 3,7,11-trimethyl- (1.11%), 2,5-Cyclohexadiene-1,4-dione, 2,6-bis(1,1-dimethylethyl)- (0.4%), and 5,9-Tetradecadiyne (0.32%). The antioxidant capacity of *GJEO* is around 1.25 µg equivalent of ascorbic acid/mL for TAC assay and IC50 = 19.05 µL/mL for DPPH test. *GJEO* exhibited significant antimicrobial activity, particularly against *Pseudomonas aeruginosa*, with a minimum inhibitory concentration (MIC) of 16.67 µL/mL. In silico molecular docking analysis revealed strong interactions between ethyleugenol characterized by multiple bonding interactions, including Pi–Alkyl and carbon–hydrogen bonds, while α-Terpineol formed hydrogen and alkyl interactions. These results underline the potential of *Gardenia jasminoides* essential oil as a promising source of bioactive compounds with antioxidant and antimicrobial properties, highlighting its possible applications in pharmaceuticals and natural therapies.

## 1. Introduction

Aromatic and medicinal plants have been integral to human civilization for thousands of years, serving not only as vital components in traditional and modern medicine but also as sources of captivating fragrances that enhance our sensory experiences. These plants, characterized by their unique ability to produce essential oils and bioactive compounds, play a crucial role in various industries including pharmaceuticals, cosmetics, and food production [1]. The diverse therapeutic properties of medicinal plants, from anti-inflammatory and antimicrobial to anticancer and antioxidant effects, have been extensively studied and utilized in the development of natural remedies and pharmaceuticals [2,3,4]. Meanwhile, aromatic plants, known for their pleasant scents and flavors, are indispensable in aromatherapy, perfumery, and culinary arts [5]. The growing interest in natural and sustainable products has spurred research into the cultivation, extraction, and application of these plants, highlighting their potential to contribute to health and well-being while also supporting ecological and economic sustainability [6]. Morocco, with its rich biodiversity and favorable climate, is home to a vast array of aromatic and medicinal plants. The country’s traditional knowledge, passed down through generations, combined with modern scientific research, has spurred significant advancements in the cultivation and utilization of these plants [7,8].

*Gardenia jasminoides* (Rubiaceae family), a flowering plant native to Asia, has long been esteemed for its ornamental beauty and aromatic allure. Beyond its visual and olfactory appeal, this botanical treasure harbors a wealth of bioactive compounds within its essential oil, holding promise for various therapeutic applications. In recent years, scientific interest has surged in unraveling the intricate chemical composition and pharmacological properties of *GJEO* [9].

The allure of *GJEO* extends far beyond its enchanting aroma. Embedded within its molecular structure are a plethora of bioactive compounds, each contributing to its distinct fragrance and potential medicinal benefits. Through meticulous analysis using state-of-the-art techniques, researchers seek to unravel the intricate chemical composition in *Gardenia jasminoides* extracts and its essential oil, shedding light on the diversity and abundance of its constituents [10]. The study carried out by [11] shows that more than 80 volatile compounds were identified among 200 compounds detected in different part of *Gardenia jasminoides*. The olfactorily valuable compounds found were less than 2.14% [11]. The major compounds detected by GC-MS in the work of Chaichana et al. [12] were linalool (21.49%), alpha-farnesene, z-3hexenyl tiglate (12.67%), and trans-beta-ocimene (8.64%). Xiao et al. [13] focused on two major phytochemicals of *GJEO*, genipin and crocin, which possess potent medicinal properties. Numerous chemical components have been identified in the fruit, including iridoids, terpenoids, flavonoids, organic acids, and volatile oils [14,15]. Song et al. [16] identified two new molecules (3beta,23- dihydroxyurs-12-en-28-oic acid and 3beta,19alpha-dihydroxy-urs-12-en-28-oic acid) among 15 compounds from flowers of *G. jasminoides* using nuclear magnetic resonance spectroscopy (NMR) techniques and physicochemical properties. Generally, study outcomes vary according to the technique used and the nature of samples and their geographical origine.

A variety of valuable phytochemical compounds have been isolated, reported to be anticancer, anti-inflammatory, antiviral, antibacterial, antidepressant, neuroprotective, and antiprotozoal. In addition, they are useful in the treatment of ankle sprains, osteoporosis and melanogenesis inhibition, osteoporosis, and inhibition of melanogenesis [17]. One of the most intriguing aspects of *GJEO* is its potential antioxidant properties. In an age where oxidative stress is implicated in numerous chronic diseases and aging processes, the quest for natural antioxidants has never been more pressing. Preliminary studies suggest that *GJEO* may harbor antioxidant compounds capable of scavenging free radicals and protecting cells from oxidative damage. Through rigorous antioxidant assays, researchers aim to elucidate the extent of its antioxidant activity and explore its therapeutic implications [13]. In addition, investigating the *G. jasminoides* extracts, Xiao et al. [13] reported that both water and ethanol extracts from fruit of *G. jasminoides* had been found to exert antioxidant activity. The water extract had a higher antioxidant activity than the ethanol extract. Moreover, *GJEO* has garnered attention for its purported antimicrobial properties. In an era marked by the rise of antibiotic resistance, the search for alternative antimicrobial agents has become increasingly urgent. Preliminary investigations suggest that *GJEO* may exhibit antimicrobial activity against a broad spectrum of pathogens, including bacteria, fungi, and viruses. However, the antibacterial activity of essential oils also varies according to the structure of bacteria, such as Gram-positive and Gram-negative bacteria, whose cell membrane compositions differ [18,19]. By subjecting the essential oil to comprehensive antimicrobial screening assays, researchers strive to elucidate its efficacy and explore its potential as a natural antimicrobial agent [20].

In addition to experimental approaches, computational techniques offer valuable insights into the pharmacological properties of *GJEO*. Through in silico evaluation, researchers can predict its interactions with biological targets, identify potential bioactive compounds, and elucidate its molecular mechanisms of action. By combining experimental and computational approaches, researchers aim to gain a comprehensive understanding of the therapeutic potential of *GJEO* and pave the way for its future applications in healthcare and biotechnology.

This scientific paper aims to explore *GJEO*, its chemical composition using GC-MS, antioxidant and antimicrobial properties, and in silico attributes.

## 2. Results and Discussion

### 2.1. Phytochemical Analysis of GJEO

The chemical composition of essential oils represents a captivating mosaic of volatile and semi-volatile organic compounds, each contributing distinct sensory and biological attributes. By employing advanced analytical techniques such as GC-MS, this research endeavors to delineate the intricate fingerprint of *GJEO*, elucidating the relative abundance and diversity of its constituents. Such knowledge forms the cornerstone for unraveling the pharmacological mechanisms and therapeutic potential of this botanical essence [9].

The GC-MS phytochemical analysis revealed the presence of 25 compounds in the volatile extract. The identified compounds are summarized in Table 1 and Figure 1. These compounds contribute to the distinctive aroma and potential bioactivity of volatile extract. In comparison with the previous studies [11,12,17,21,22,23], our outcome showed five new compounds: Methyleugenol (15.41%), 1-Undecyne (3.4%), 2,6,10-Dodecatrien-1-ol, 3,7,11-trimethyl- (1.11%), 2,5-Cyclohexadiene-1,4-dione, 2,6-bis(1,1-dimethylethyl)- (0.4%), and 5,9-Tetradecadiyne (0.32%). These compounds make up approximately 21.56% of the *GJEO*. In addition, among the detected compounds, there are Methyleugenol (15.41%); Tricyclo[2.2.1.0(2,6)]heptane, 1,3,3-trimethyl- (10.68%); 2-Methyl-1-phenyl-1-butanol (8.25%), 2-Butanone, 4-(4-methoxyphenyl)- (6.64%); and Eucalyptol (4.13%). These major compounds have been identified to have different effects on the human body, such as anticancer, anti-inflammatory, antioxidant, and antimicrobial activities. Methyleugenol or eugenol methyl, derived from eugenol, known by the following synonyms, 4-allylveratrole; 4-allyl-1,2-dimethoxybenzene; eugenyl methyl ether; 1,2-dimethoxy-4-(2-propenyl)benzene; 3,4-dimethoxy-allylbenzene; 3-(3,4-dimethoxyphenyl)prop-l-ene; O-methyleugenol; and methyl eugenol ether, is one of the natural products extracted from several aromatic plants [24,25,26]. It is known for its use in foodstuffs as flavoring agents. According to [27], the exposure to methyl eugenol resulting from consumption of foods containing these molecules does not present a significant cancer risk. However, it is a genotoxic carcinogen with DNA-binding capacity [24]. Tricyclo[2.2.1.0(2,6)]heptane, 1,3,3-trimethyl-, also known as Cyclofenchen; Tricyclo[2.2.1.02,6]heptane, 1,3,3-trimethyl-; or 1,3,3-Trimethyltricyclo[2.2.1.02,6]heptane, is a substance identified in some plant materials such as *Viguiera dentata* (30.85%) [28], *Ziziphora clinopodioides* subsp. *rigida* (25.29%) [29], *Helichrysum italicum* (14.20%), *Cymbopogon martini* (13.91%) [30], *Salvia tomentosa* (10%) [31], and *Eucalyptus maculata* (7%) [32]. Its biological activities are not widely recognized or established in the scientific literature. 2-Methyl-1-phenyl-1-butanol, also named 2-Methyl-1-phenyl-1-butanol or 2-methyl-1-phenylbutan-1-ol, is one of the compounds identified in *Baccopa monnieri* (2.5%) [33]. Also, this study revealed the presence of 4-(4-Methoxyphenyl)-2-butanone or Anisylacetone, which play an important role as antimicrobial agents [34]. Benzoic acid, 2,6-dimethoxy-, methyl ester, also named Methyl 2,6-dimethoxybenzoate, 2,6-Dihydroxybenzoic acid, dimethyl ether, or methyl ester, is a derived compound of benzoic acid, naturally found in *Dianthus caryophyllus*, *Curculigo orchioides*, *Molineria latifolia*, and *Molineria capitulate* [35,36]. Several studies have demonstrated the great biological effect of hydroxybenzoic acids derived thanks to their geometry, the electronic charge distribution in their molecules, their acidity, and their lipophilicity [37,38]. Eucalyptol or cineole is a terpene derivative traditionally used as a respiratory tract antiseptic [39]. It is used in mouthwashes, in insect repellents, and as a cough suppressant [40]. It is considered a major component of eucalyptus oil, and naturally found in several plant species such as *Curcuma xanthorrhiza, Baeckea frutescens, Peumus boldus* leaf, and *Paeonia lactiflora* root [41]. In addition, other important compounds were identified at low percentages, such as Eugenol (3.12%), α-Terpineol (3.37%), Isoborneol (0.42%), and Linalool (0.59%), revealing many biological effects. These compounds are widely used in natural preservatives, antiseptics, and essential oil-based antimicrobial treatments.

In comparison with earlier studies, a study carried out by [12] reported a diverse range of compounds in the essential oil of the same species studied. Their findings corroborate the presence of various major compounds such as Linalool (21.49%), α-Farnesene (17.54%), z-3-hexenyl tiglate (12.67%), and trans-β-Ocimene (8.64%) with one shared compound, Linalool (0.59%). Findings of [11] reported the presence of numerous compounds, some of which are common with our findings, such as eugenol, Limonene, Eucalyptol (1,8 Cineole), Linalool, and α-Tetpineol. In addition, the study of [23] indicated the presence of Linalool, α-Terpineol, Tricosane, and Pentacosane among 18 components in the essential oils of *Gardenia jasminoides* J. Ellis and *Gardenia jasminoides* f. *longicarpa* flowers.

### 2.2. In Vitro Antioxidant Activity of GJEO

Antioxidant activity is an essential feature in the search for natural remedies for conditions linked to oxidative stress. Free radicals, generated during metabolic processes or environmental exposures, pose a constant threat to cellular integrity, contributing to aging, inflammation, and various chronic diseases. Here, we explore the antioxidant prowess of *GJEO*, seeking to elucidate its ability to scavenge free radicals and mitigate oxidative damage [42]. The outcome presented in Table 2 below shows that *GJEO* has remarkable antioxidant activity. The value of DPPH-IC50 of *GJEO* (19.05 µL/mL) is higher than that of the ascorbic acid (7.48 µg/mL) and BHT (3.12 µg/mL).

### 2.3. Antimicrobial Analysis

The results presented in Table 3 represent antimicrobial activity using zone of inhibition, MIC, and MBC. The results obtained for antibacterial activity using the diameter of the zone of inhibition show that *GJEO* has moderate antibacterial activity against the studied bacterial strains, with inhibition zones ranging from 10.67 to 14.33 mm. However, its effectiveness is lower compared to Ampicillin, which has a much larger inhibition zone (23 mm) against *E. coli*. Penicillin also shows antibacterial activity but is less effective than *GJEO* against *S. aureus* and has a smaller inhibition zone against *E. coli* (12 mm vs. *GJEO*’s 10.67 mm). Comparing the minimum inhibitory concentration (MIC) values, the *P. aeruginosa* has the lowest MIC at 16.67 µg/mL, indicating that *GJEO* is most effective at inhibiting the growth of this bacterium and can kill it at MBC of 40 µg/mL. *E. coli* and *S. aureus* both have a higher MIC of 26.67 µg/mL, suggesting that *GJEO* is less effective against these bacteria compared to *P. aeruginosa*. In terms of antifungal activity, *GJEO* is ineffective against *C. albicans* and *A. niger*.

The antimicrobial properties of the *GJEO* are due to its wealth of bioactive molecules. Indeed, according to several studies, Eugenol, Thymoquinone, Methyleugenol, Nerolidol, and α-Terpineol show strong antimicrobial properties against a broad spectrum of bacteria and fungi. The study carried out by [43] on the Eugenol component indicates its antimicrobial effects on strains responsible for human infectious diseases, diseases of the oral cavity, and foodborne pathogens, as well as those with multidrug resistance. According to several studies, Eugenol has an antibacterial effect against *E. coli* at (MIC and MBC = 1600 µg/mL) [44]; *S. aureus* ATCC 25923 (MIC and MBC = 0.4 µL/mL) [45]; *P. fluorescens* ATCC 13525 (DIZ = 14.8 mm) [46]; *C. albicans* MIC: 625 µg/mL, MFC: 1209 µg/mL. According to Hu, Zhou, and Wei (2018), Eugenol (at 0.01%, *V*/*V*) can reduce the production of pyocyanin and 2-heptyl-3-hydroxy-4(1H)-quinolone and inhibit the swarming motility and hemolytic activity of *P. aeruginosa.* Eugenol (0.80 mM) expresses no effect on the growth of *Aspergillus flavus*, but completely inhibits the aflatoxin B1 (AF B1) biosynthesis production with a inhibition rate of 95.4% [47]. In addition, the presence of some substances such as ricyclo[2.2.1.0(2,6)]heptane, 1,3,3-trimethyl, 4-(4-Methoxyphenyl)-2-butanone (Anisylacetone) in the sample can behave as an antimicrobial substance [31,34,48,49]. Essential oils and plant-derived natural products are increasingly studied as alternative strategies against multidrug-resistant (MDR) bacteria due to their antimicrobial properties. Targeting quorum sensing mechanisms (QSMs) and biofilm formation is crucial in combating chronic infections, with research focusing on quorum sensing inhibitors (QSI) as novel antimicrobial agents [50]. This aligns with Hetta et al. [51], who highlight QSIs as promising alternatives for MDR bacterial control. Different mechanisms are proposed for the antibacterial activity of essential oils. Essential oils exert antibacterial activity by disrupting bacterial cell membranes, increasing their permeability and causing cellular components to leak out. Their lipophilic nature enables them to penetrate membranes, affecting both the outer envelope and the cytoplasm. This disruption leads to reduced membrane potential, altered proton pumps, ATP depletion, and impaired cellular respiration. In addition, essential oils interfere with fatty acids, phospholipid bilayers, and polysaccharides, leading to cytoplasmic coagulation and destabilization of lipid–protein interactions. These combined effects ultimately inhibit bacterial growth and viability [52].

### 2.4. In Silico Evaluation of GJEO

In silico evaluation, leveraging the power of computational modeling and simulation, augments our understanding of molecular interactions and biological activities. By employing cutting-edge computational techniques, we aim to unravel the intricate interplay between *GJEO* constituents and their putative biological targets. Through in silico exploration, we aspire to forecast the pharmacokinetic properties, target engagement, and therapeutic potential of this botanical elixir, guiding future experimental endeavors and therapeutic applications [53].

In this study, we determined the potential antimicrobial activity of Eugenol, Methyleugenol, and α-Terpineol ligands. These compounds used in our study had the highest percentages, respectively, 3.37%, 3.12%, and 15.41%. The interaction types between these compounds and topoisomerase II receptors were determined by molecular docking. The results of the 3D (three-dimensional) and 2D (two-dimensional) view are listed in Table 4.

The results show that none of the compounds had unfavorable interactions. The interaction types of Eugenol with the topoisomerase II active site consist of three carbon hydrogen bonds with Gln190, Val11, and Tyr354; one conventional hydrogen bond with Gln174; and one Alkyl type with Leu70. Methyleugenol has three different interaction types with the topoisomerase II active site, including two carbon–hydrogen bonds with Asp177 and Asp40, and one Pi–Alkyl interaction with Tyr36. Moreover, the extracted compound α-Terpineol has one conventional hydrogen bond with Thr75 and two interactions that are a mix of Alkyl and Pi–Alkyl types with Tyr36 and Cys37. Several studies indicate that ciprofloxacin binds to human topoisomerase II primarily through hydrogen bonds, notably with Asp479 and Ser480. Additionally, π-π stacking and electrostatic interactions contribute to the stability of the ciprofloxacin–topoisomerase II complex. Eugenol compound was acknowledged with a maximum total score of 5.68, followed by a total score of 5.01 for α-Terpineol compound; among the selected docked compounds, compound Methyleugenol was recognized with the lowest total score of 4.84.

The human protein 3MNG is a target for a study of antioxidant activity, and the extracted compounds (Eugenol, Methyleugenol, and α-Terpineol were therefore subjected to a selective Peroxiredoxins protein (PDB: 3MNG) active for their molecular docking study. The results of the 3D and 2D view are listed in Table 5.

Eugenol exhibits one Pi–Alkyl interaction with Arg95, one Pi-Anion interaction with Glu16, and three carbon–hydrogen bond interactions with Arg95, Ala90, and Val94. This compound is also characterized by an unfavorable donor–donor interaction with Leu96, which decreases the stability of Eugenol. The hydroxyl group might facilitate interactions with the active cysteine residues of Peroxiredoxin, leading to a reasonable binding affinity. Methyleugenol has two types of Pi–Alkyl interactions: Ala90 and Arg95; three carbon-hydrogen bond interactions: Gly85, Gly82, and Thr81; and one Pi-Sigma interaction: Leu96. Moreover, α-Terpineol has one conventional hydrogen bond with Glu16 and two Alkyl interactions with Ala90 and Arg86. Its structural flexibility and alcohol functionality could contribute to interactions but may not be as strong as Eugenol. The α-Terpineol compound was acknowledged with a maximum total score of 5.12, followed by a total score of 4.48 for Methyleugenol compound; among the selected docked compounds, compound Eugenol was recognized with the lowest total score of 3.86.

## 3. Materials and Methods

### 3.1. Plant Material and Essential Oil Preparation

The flowers of (*G. jasminoides*) were collected in May 2022 in El Jadida province, Morocco. The *G. jasminoides* essential oil (*GJEO*) was extracted from the fresh flower by hydro-distillation for 4 h. The upper essential oil in the oil–water separator was collected. From 1 kg of fresh flowers, 0.07 mL of essential oil could be extracted on average.

### 3.2. Identification of the Constituents of Essential Oil Using GC-MS

The crude extract’s moderately polar fractions displayed significant anticandidal properties, prompting us to use GC-MS analysis for the purpose of characterizing their chemical compositions and detecting compounds concealed within [9].

The essential oil was subjected to derivatization through overnight incubation with a silanizing reagent (HMDS/TMCS/pyridine in a 3:1:9 *v*/*v*/*v* ratio) at room temperature to produce trimethyl silyl (TMS) derivatives, before being analyzed using GC-MS. For GC-MS analysis, a gas chromatograph coupled to a mass spectrophotometer (Brand Agilent Technologies Model 5973) was employed with a capillary column with model number Agilent 19091S-433, 30.0 m in length, 250 μm in diameter, and a film thickness of 0.25 μm. The GC was equipped with an electronically controlled split/splitless injection port. The injection volume was 1 µL at a temperature of 260 °C. The oven temperature followed this program: it started at 60 °C, ramped up to 300 °C over 10 min, and then remained at 300 °C for 20 min. The detector was set at 250 °C, and helium was used as the carrier gas with a total flow rate of 31.4 mL/min, and the split ratio was 30:1. The silylated compounds were analyzed using the GC-MS Wiley 7n.l Library.

### 3.3. In Vitro Determination of Antioxidant Activity of GJEO

Determination of the antioxidant activity of *GJEO* was carried out by the DPPH and TAC method described in our previous publication [54], with a slight modification, using an essential oil concentration of 100 µL/mL.

### 3.4. Antimicrobial Analysis Using Disk Diffusion Assay

This study utilized five microbial strains, including three bacterial strains (*Escherichia coli* ATCC 25922, *Pseudomonas aeruginosa*, and *Staphylococcus aureus* ATCC 29213) and two fungal strains (*Candida albicans* ATCC 10231 and *Aspergillus niger* ATCC 16404). Standardization and inoculation were performed using established methods for bacteria and fungi.

The disk diffusion assay (Kirby–Bauer method) was employed to evaluate antimicrobial activity. Standardized suspensions (1–5 × 10^8^ CFU/mL) were inoculated onto Mueller–Hinton agar, and Whatman paper discs (6 mm) impregnated with 20 µL of *GJEO* were placed on the agar surface. Plates were incubated at 37 °C for 24 h, and inhibition zone diameters were measured in millimeters. Positive controls included Ampicillin (10 µg/disc), Penicillin (10 µg/disc), Fluconazole (25 µg/disc), and Voriconazole (50 µg/disc).

### 3.5. Determination of the Minimum Inhibitory Concentration (MIC)

The minimum inhibitory concentration (MIC) of the *GJEO* samples was determined using microdilution assays national committee for clinical laboratory standards (NCCLS). Ten concentrations (500 to 3.91 µL/mL) were prepared by two-fold serial dilutions in dimethylsulfoxide (DMSO). In sterile conditions, 50 µL of Mueller–Hinton broth was added to each well of a microplate, with the first well as a negative control (100 µL of essential oil) and the last well as a growth control. Microdilutions were performed by transferring 50 µL across the wells, followed by the addition of 50 µL of microbial suspension to each well.

The plates were incubated for 24 h at 37 °C for bacteria and 25 °C for fungi. Results were read using a colorimetric method with 2,3,5-triphenyltetrazolium chloride (TTC). After 2 h of additional incubation, MIC was defined as the lowest concentration of *GJEO* that inhibited microbial growth, indicated by the absence of a pinkish coloration due to dehydrogenase activity.

### 3.6. Determination of the Minimum Fungicidal/Bactericidal Concentration

Minimum fungicidal/bactericidal concentration was determined using a cotton swab; three wells were sampled, adjacent to the MIC wells. After their growth on the surface of the non-selective agar plate, the surviving cell number (CFU/mL) was determined after 24 h of incubation. The bactericidal endpoint (BEM) was determined as the lowest concentration at which approximately 99.9% of the inoculum was killed [55].

### 3.7. Molecular Docking

Molecular docking is an established silico structure-based method widely used in drug discovery; it is among the most successful and popular structure-based in silico methods, which help predict the interactions occurring between compounds and biological targets. Molecular docking enables the identification of compounds of therapeutic interest, predicting ligand–target interactions at a molecular level [56,57]. The activities (antimicrobial, antioxidant), and structures of compounds (Eugenol, Methyleugenol, and α-Terpineol) are shown in Table 6.

This study begins by selecting the target proteins and downloading them from the Protein Data Bank (RCSB PDB) [58]; the preparation of these proteins begins with elimination of water molecules by using discovery software [59]. After determining the active site, the ligand and the protein are transformed into PDB files using sybyl-X 1.3 software [60], and both files are compressed into a single file using the pymol application [61].

Using in silico molecular docking, we selected antimicrobial and antioxidant activities; we used topoisomerase II receptors (PDB ID: 1JIJ) and human Peroxiredoxins (PDB: 3MNG), respectively.

### 3.8. Statistical Analysis

The data’s mean and standard deviation (SD) were determined. One-way analysis of variance was used to examine the data. Different letters indicate a significant difference between means (*p* < 0.05).

## 4. Conclusions

This study analyzed the chemical composition, antioxidant activity, antimicrobial properties, and molecular interactions of *Gardenia jasminoides* essential oil (*GJEO*). About 25 volatile compounds were identified, including 5 new detected compounds, using GC-MS. These compounds exhibit various biological activities, including antioxidant, anticancer, anti-inflammatory, and antimicrobial effects. Antioxidant evaluation demonstrated strong free radical scavenging activity. Antimicrobial testing showed moderate antibacterial activity against studied bacterial strains, and no antifungal activity. The antimicrobial effects were attributed to compounds like Eugenol, Methyleugenol, and α-Terpineol. In silico molecular docking revealed interactions of Eugenol, Methyleugenol, and α-Terpineol with key proteins (topoisomerase II and peroxiredoxins), showing hydrogen bonds and Pi–Alkyl interactions. While Eugenol exhibited reduced stability due to unfavorable interactions, all compounds showed promising potential for therapeutic applications. Overall, *GJEO* demonstrates significant antioxidant properties and moderate antimicrobial activity, supported by its rich phytochemical profile and computational analyses.

## Figures and Tables

**Figure 1 plants-14-01055-f001:**
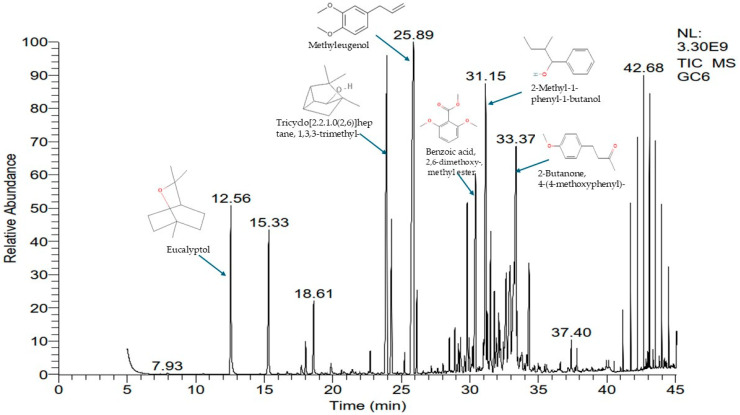
GC-MS chromatogram of the *GJEO*. The numbers on the peaks indicate the retention times of the various compounds.

**Table 1 plants-14-01055-t001:** Identified compounds in *GJEO* through GC-MS.

RT (min)	Area %	Compound	MF	Prob
12.56	4.13	Eucalyptol	C10H18O	87.63
15.33	3.37	D-Limonene	C10H16	13.42
18.02	0.59	Linalool	C10H18O	46.72
18.60	1.42	α-Terpineol	C10H18O	89.57
23.95	10.68	Tricyclo[2.2.1.0(2,6)]heptane, 1,3,3-trimethyl-	C10H16	39.75
24.29	3.12	Eugenol	C10H12O2	73.84
25.23	0.32	5,9-Tetradecadiyne	C14H22	100
25.88	15.41	Methyleugenol	C11H14O2	82.28
26.13	1.16	α-Pinene	C10H16	2.97
29.81	3.4	1-Undecyne	C11H20	25.49
30.21	0.56	Limonene	C10H16	33.94
31	0.47	2,6,10-Dodecatrien-1-ol, 3,7,11-trimethyl-	C15H26O	99
31.15	8.25	2-Methyl-1-phenyl-1-butanol	C11H16O	100
31.27	0.83	3-Buten-2-one, 4-(2,6,6-trimethyl-1-cyclohexen-1-yl)-	C13H20O	89.44
31.51	2.63	2,3-Dimethoxybenzoic acid	C9H10O4	15.3
31.78	1.23	Nerolidol	C15H26O	91.29
32.1	0.98	Benzene, 2-methyl-1,4-bis(1-methylethyl)-	C13H20	67.91
32.19	1.11	Supraene	C15H26O	86.33
32.44	0.42	Isoborneol	C10H18O	63.77
32.64	2.87	Benzene, 1-(1,1-dimethylethyl)-3-methyl-	C11H16	17.22
33.36	6.64	2-Butanone, 4-(4-methoxyphenyl)-	C11H14O2	43.79
33.79	0.57	Bicyclo[3.1.1]hept-2-ene-2-ethanol, 6,6-dimethyl-	C11H18O	28.08
34.15	0.4	2,5-Cyclohexadiene-1,4-dione, 2,6-bis(1,1-dimethylethyl)-	C14H20O2	80.01
37.4	0.56	Undec-10-ynoic acid	C11H18O2	28.63
37.82	0.36	Thymoquinone	C10H12O2	31.94

RT (Retention Time), Prob (Probability), MF (Molecular formula).

**Table 2 plants-14-01055-t002:** DPPH radical scavenging activity and total antioxidant activity.

	DPPH-IC50 (µL/mL)	TAC (µg EAA/mL)
*GJEO*	19.05 ± 1.87 c	1.247 ± 0.06
BHT *	3.12 ± 0.32 a	-
Ascorbic acid *	7.48 ± 0.02 b	-

DPPH: 2,2-Diphenyl-1-picrylhydrazyl; IC50: half-maximal inhibitory concentration; TAC: total antioxidant capacity; *GJEO*: *Gardenia jasminoides essential oil*; BHT: butylated hydroxytoluene. * unit used for DPPH-IC50 is (µg/mL). (-): not determined. (a–c) indicates significant difference between means at *p* < 0.05.

**Table 3 plants-14-01055-t003:** Antimicrobial activity of the *GJEO* against various pathogens.

	*E. coli*			*P. aeruginosa*			*S. aureus*			*C. albicans*			*A. neger*		
	DIZ	MIC *	MBC *	DIZ	MIC	MBC	DIZ	MIC	MBC	DIZ	MIC	MFC	DIZ	MIC	MFC
*GJEO* (100 µL/mL)	10.67 ± 0.58	26.67 ± 11.55	40 ± 0.00	11.00 ± 0.00	16.67 ± 5.77	20 ± 0.00	14.33 ± 0.58	26.67 ± 11.55	-	Rs	Rs	Rs	Rs	Rs	Rs
Ampicillin (10 µg/disc)	23.00 ± 0.00	-	-	Rs	-	-	15.00 ± 0.00	-	-	-	-	-	-	-	-
Penicillin (10 µg/disc)	12.00 ± 0.00	-	-	Rs	-	-	Rs	-	-	-	-	-	-	-	-
Fluconazole (25 µg/disc)	-	-	-	-	-	-	-	-	-	21.00 ± 0.00	0.40 ± 0.00	-	-	-	-
Voriconazole (50 µg/disc)	-	-	-	-	-	-	-	-	-	-	-	-	12.00 ± 0.10	0.5 ± 0.00	-

DIZ: diameter of inhibition zone (mm); MIC: minimum inhibitory concentration; MBC: minimum bactericidal concentration; MFC: minimum fungicidal concentration. * MIC and MBC expressed in (µL/mL) for *GJEO* and (mg/mL) for antibiotics; Rs: resistant. (-): not determined.

**Table 4 plants-14-01055-t004:** Three- and two-dimensional view of the binding conformation for the compounds (Eugenol, Methyleugenol, and α-Terpineol) with topoisomerase II complex (PDB: 1JIJ) and docking score.

N°	3D View	2D View
Eugenol	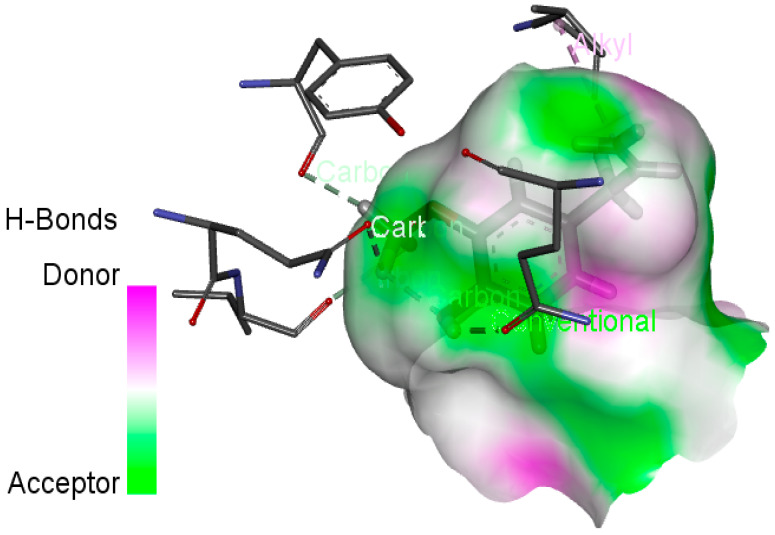	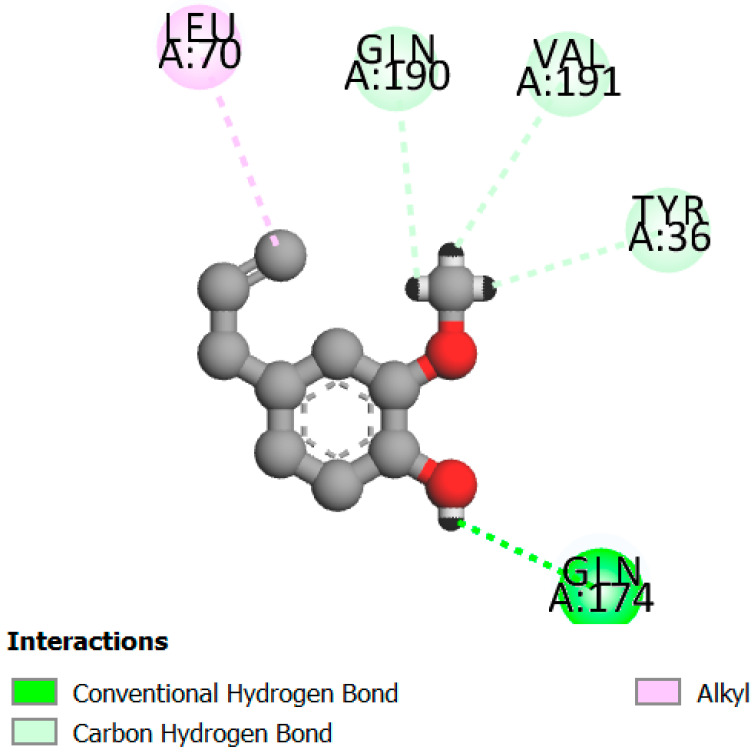
Methyleugenol	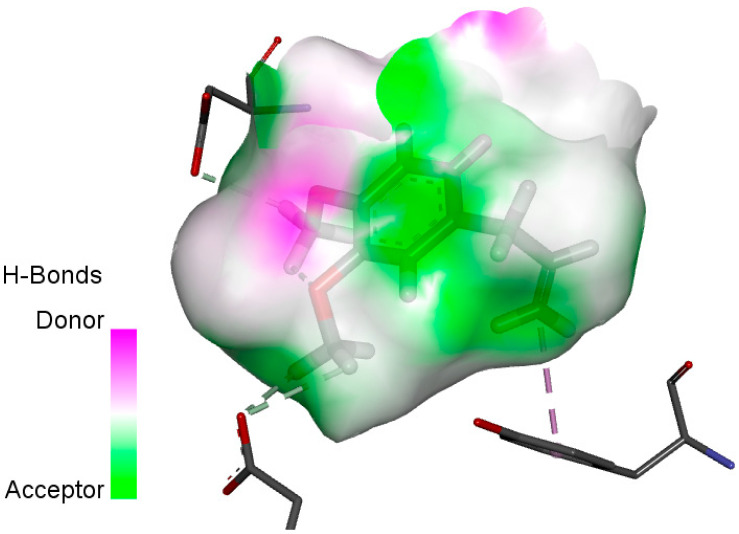	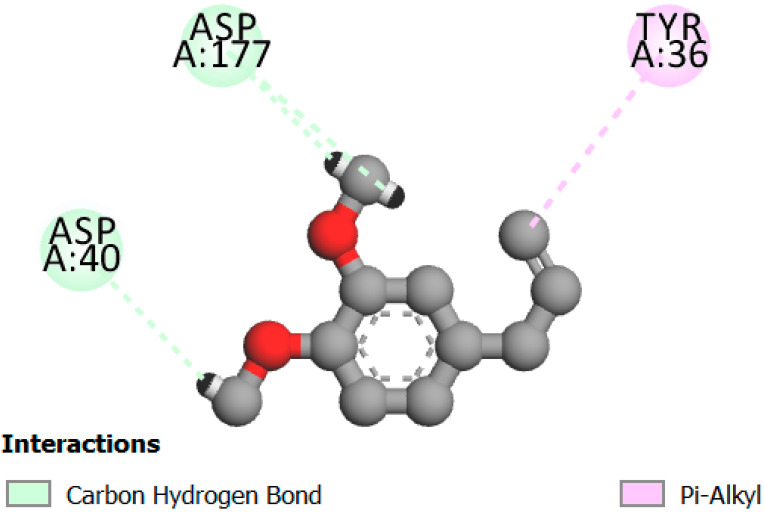
α-Terpineol	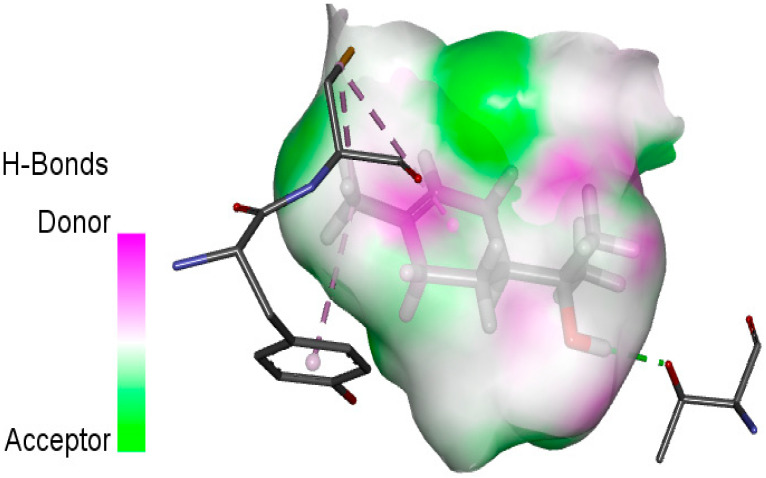	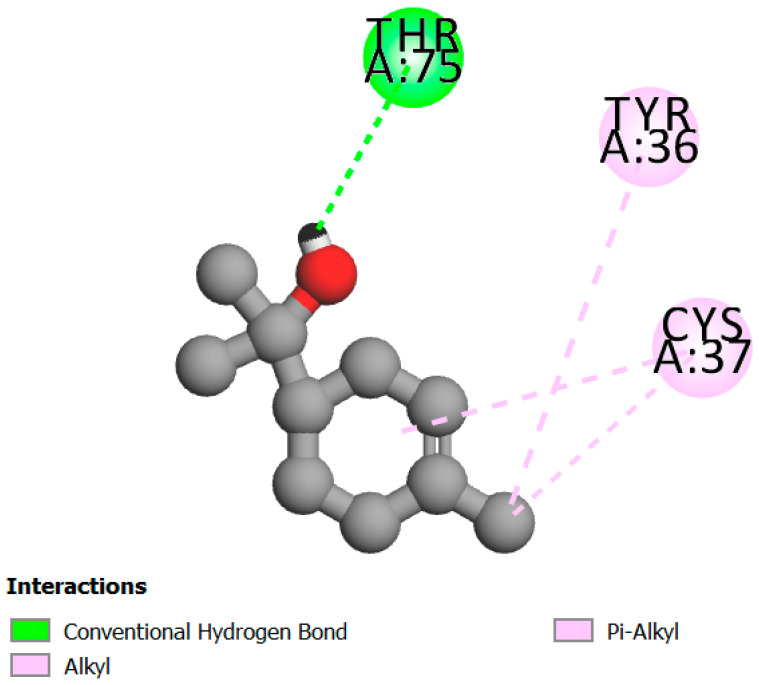

**Table 5 plants-14-01055-t005:** Three- and two-dimensional view of the binding conformation for the compounds (Eugenol, Methyleugenol, and α-Terpineol) with Peroxiredoxins (PDB: 3MNG) and docking score.

N°	3D View	2D View
Eugenol	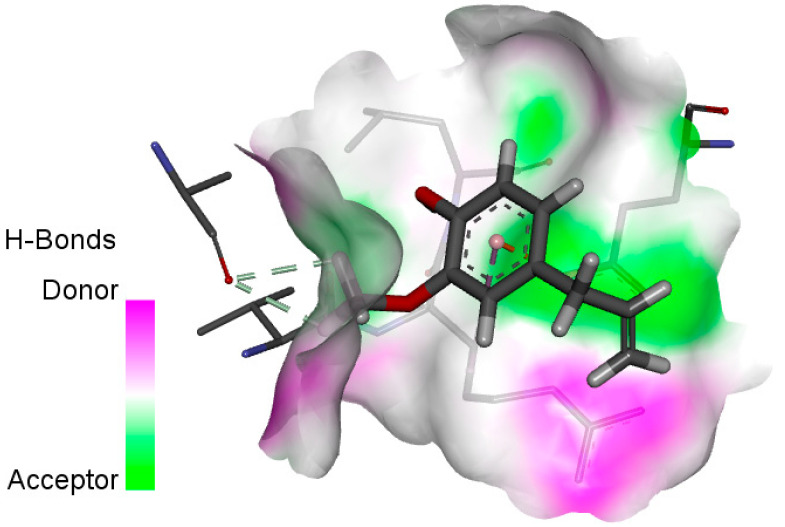	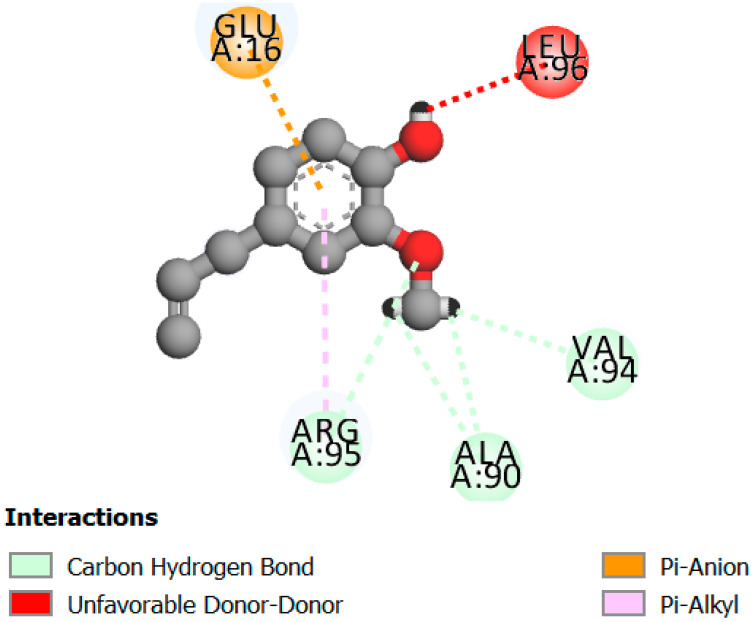
Methyleugenol	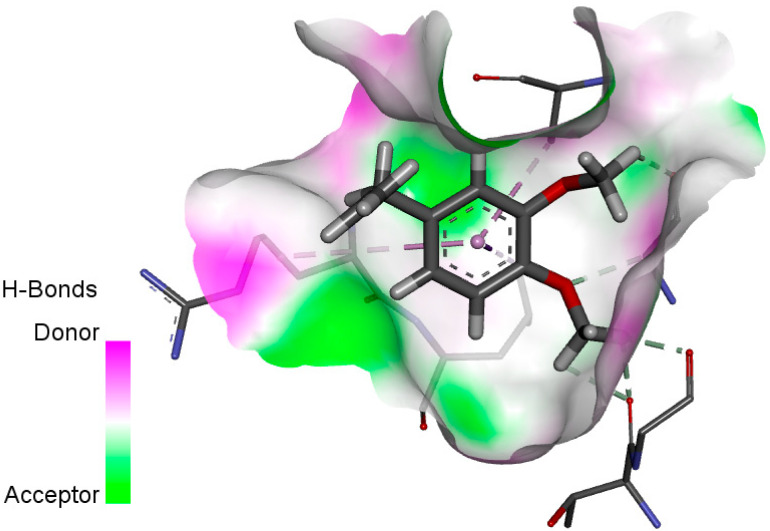	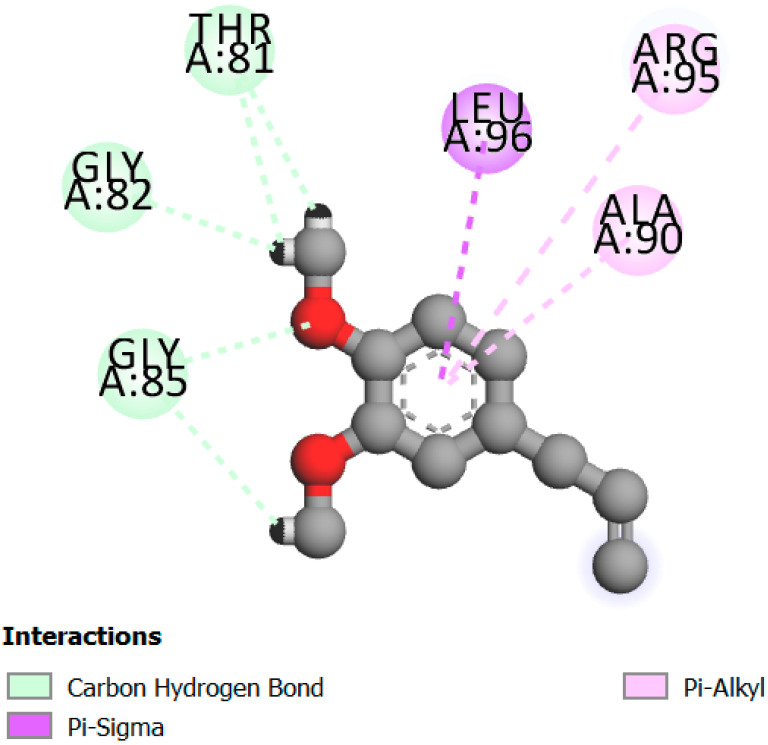
α-Terpineol	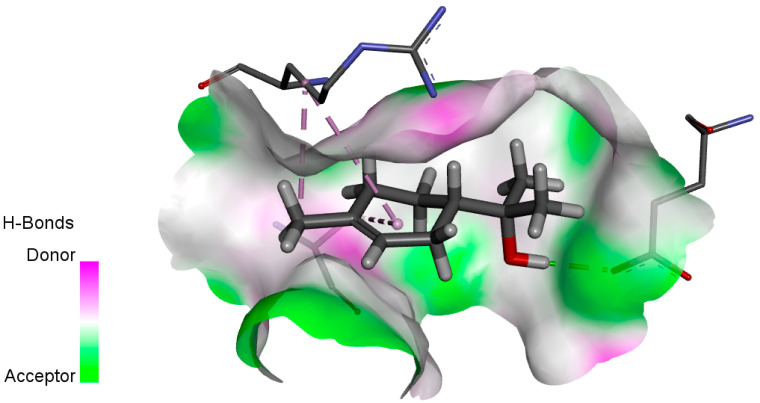	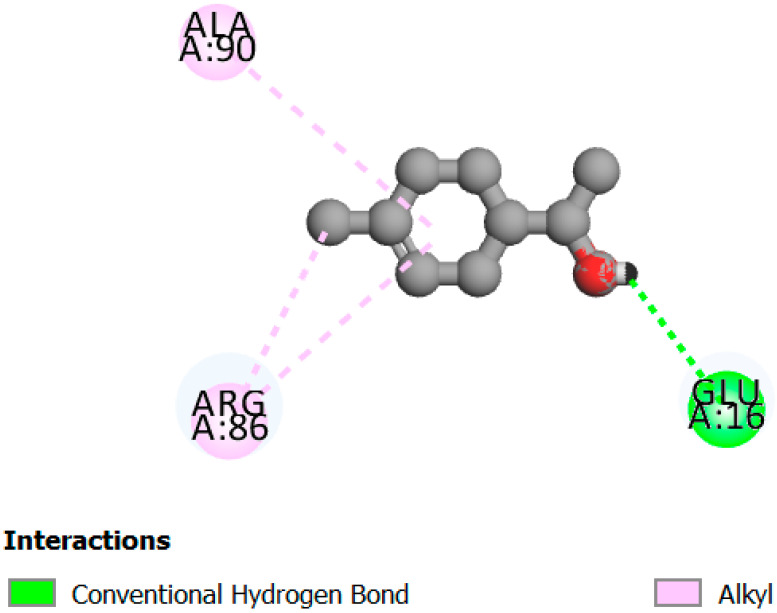

**Table 6 plants-14-01055-t006:** Structures of compounds (Eugenol, Methyleugenol, and α-Terpineol) and the protein responsible for each activity.

Compounds	Eugenol	Methyleugenol	α-Terpineol
Molecular Formula	C_10_H_12_O_2_	C_11_H_14_O_2_	C_10_H_18_O
CID	3314	7127	443162
Structures	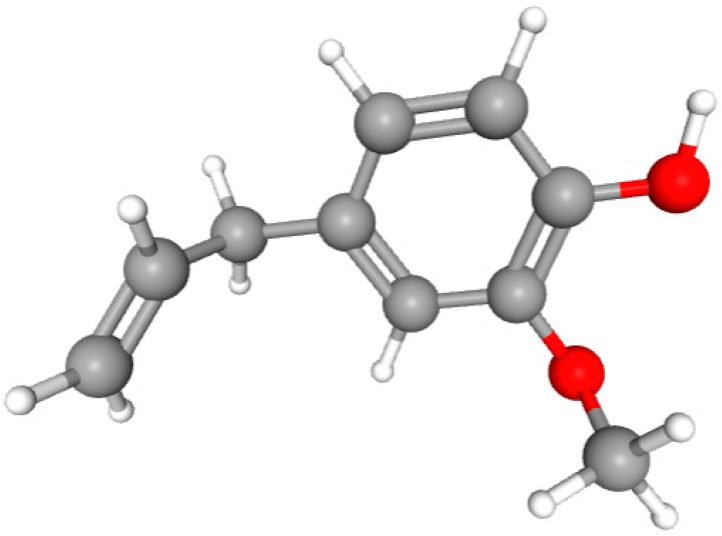	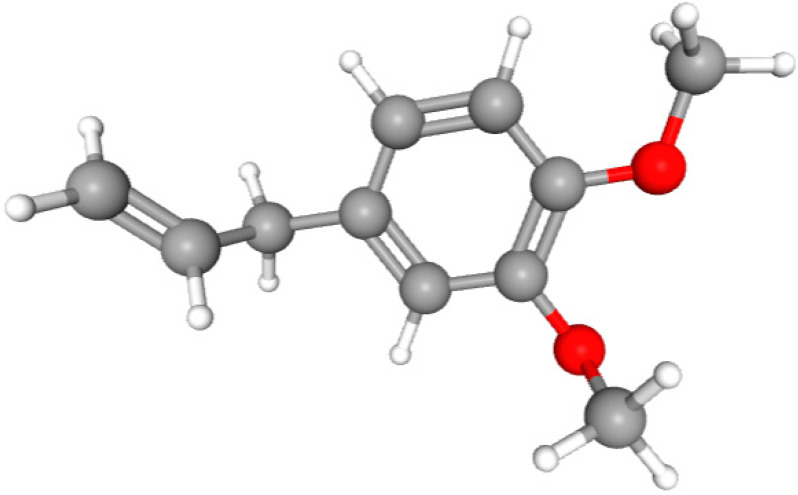	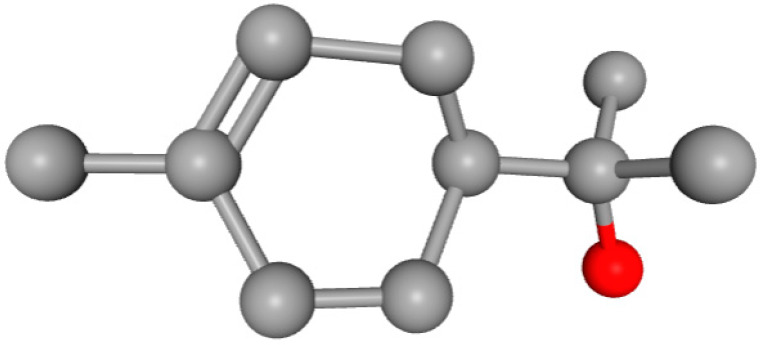
Activities	Protein		
Antimicrobial	PDB: 1JIJ	PDB: 1JIJ	PDB: 1JIJ
Antioxidant	PDB: 3MNG)	PDB: 3MNG)	PDB: 3MNG

## Data Availability

Data is contained within the article.

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
