# Peer review of "Chemical Profiling, Antioxidant and Antimicrobial Activities, and In Silico Evaluation of Gardenia jasminoides Essential Oil"

_plants, 2025, doi:10.3390/plants14071055_

Round 1
Reviewer 1 Report
Comments and Suggestions for Authors
Main corrections:
Abstract section:
The conclusions are too generic. Please improve this part.
Results and discussion:
In docking studies, please give an indication of the putative affinity of the studied compounds towards the selected targets. Additionally, try to establish if the putative affinity is consistent, at least in part, with the levels of the phytochemicals in the EO.
In table 3, please include S.D. for each treatment, including reference antimicrobial drugs.
In table 2, different letters should indicate different levels of statistical significance. However, in the caption it is not included any P value. Additionally, a statistical analysis paragraph is missing in materials and methods.
Minor corrections:
Please explain the use of abbreviations at their first appearance in the text.
Please check the use of capital and uppercase/lowercase letters.
Please check also the use of italics: sometimes it is reported GJEO and other times GJEO. Please make uniform the style.
Please check typos: i.e. line 85.
The statements section is not completed after the conclusions paragraph.
Author Response
We would like to express our gratitude to the reviewers’ for their constructive comments and valuable suggestions. Their feedback has greatly contributed to improving the quality of this manuscript. We have taken their recommendations into account and made the necessary revisions accordingly.
The revised parts are highlighted in yellow.
Main corrections:
Abstract section:
Comment 1: The conclusions are too generic. Please improve this part.
Reply 1: Thank you for your comment. I have revised and improved the abstract.
Results and discussion:
Comment 2: In docking studies, please give an indication of the putative affinity of the studied compounds towards the selected targets. Additionally, try to establish if the putative affinity is consistent, at least in part, with the levels of the phytochemicals in the EO.
Reply 2: The docking studies, based on the main results, are those obtained from the GCMS-analysis, the compounds used α-Terpineol, Eugenol, and Methyleugenol used in this study have the highest percentage area.
I added a paragraphe in 2.6. In Silico evaluation of GJEO
Comment 3: In table 3, please include S.D. for each treatment, including reference antimicrobial drugs.
Reply 3: thank you for your remark. I considered your comment.
Comment 4: In table 2, different letters should indicate different levels of statistical significance. However, in the caption it is not included any P value. Additionally, a statistical analysis paragraph is missing in materials and methods.
Reply 4: I added the missing information as required.
Minor corrections:
Comment 5: Please explain the use of abbreviations at their first appearance in the text.
Reply 5: Done as suggested.
Comment 6: Please check the use of capital and uppercase/lowercase letters.
Reply 6: I have revised this issue.
Comment 7: Please check also the use of italics: sometimes it is reported GJEO and other times GJEO. Please make uniform the style.
Reply 7: we considered your notice.
Comment 8: Please check typos: i.e. line 85.
Reply 8: Checked.
Comment 9: The statements section is not completed after the conclusions paragraph.
Reply 9: Done.
Reviewer 2 Report
Comments and Suggestions for Authors
The manuscript content would fit with the scope of Plants, but there is little new contribution in this work, and the presentation of the results is plain.
The main results are those obtained from the GCMS-analysis. It is not clear which compounds that are new, and the reliance of those identities, based on the present MS-resolution, would be low. The antioxidant part gives only one value for each of the two assays, and the antimicrobial part is also limited in results. I am not qualified to judge the contribution of the in silico part of the work.
Other comments:
The title seem to promise more than it keeps, and if published anywhere I would recommend to remove the word ‘new’.
Affiliation addresses are uncomplete.
Abstract:
Lines 29-31: This sentence is unclear. Do you mean: “The obtained results highlight the presence of 41 volatile compounds including two hitherto unreported compounds from this species: methyleugenol (15.41% of total volatiles), tricy-30 clo[2.2.1.0(2,6)]heptane, 1,3,3-trimethyl (10.68%).” Is this correct? If so, try to highlight what is the new contribution with this paper.
What is meant by TAC? Write out the full name when this shortcut occurs for the first time in the manuscript. And, what is meant by EAA?
Use only significant numbers in all given results. It is odd to report a DPPH value at 1.247±0.056, IC50=19.05±1.87 and MIC at 16.67±5.77. I guess the uncertainty in these numbers is quite high?!!
It would be interesting to know in the abstract which microbe-strains were included in the tests.
Lines 34-35: Unclear sentence: “The in-silico analysis revealed the presence of Compound Methyleugenol has two types of Pi-Alkyl…”
Lines 38-39: Remove this conclusion: “This study reveals some interesting results concerning the phytochemicals contained in GJEO and their effects on health.”
Introduction:
This is not in accordance to what would be expected from a scientific article: A brief introduction to the actual area (plant background), an update to state-of-the-art (what is the present knowledge), gaps in knowledge, and a strategy of how to fill these gaps (motivation). In the present form it is a wordiness introduction. Simply too much verbiage.
Results and discussion:
The first paragraph contains no results! Remove!
Table 1 contain most of the data in this work, but based only on one analysis(!) The title claims ‘bioactive compound’, but there are no results that verifies that the single compounds are bioactive. The MS-data are of low-resolution, and doesn’t meet the criteria to claim new identities from this plant. There is no explanation of the ranges of SI and RSI, so the reader doesn’t necessarily know what a good match would look like. Beneath the table the is an explanation to Prob. Where is Prob?
All numbers with respect to antioxidant measures as well as those of the antimicrobial analyses operates with numbers with many decimal digits. This is not good. Use only significant numbers!
What is the news in 2.3?
Line 238: What is meant by a ‘good human protein’?
References: Should be made in accordance to magazine’s instruction and guideline.
Too much self-citation. Nine!!
Comments on the Quality of English LanguageThe introduction is characterized by wordiness and less content. In some cases in the Results and Discussion section it is hard to understand the meaning (e.i. line 156, 259-261).
Author Response
We would like to express our gratitude to the reviewers’ for their constructive comments and valuable suggestions. Their feedback has greatly contributed to improving the quality of this manuscript. We have taken their recommendations into account and made the necessary revisions accordingly.
The revised parts are highlighted in yellow.
Comment 1: The manuscript content would fit with the scope of Plants, but there is little new contribution in this work, and the presentation of the results is plain.
Reply 1: Thank you for your kind comment.
Comment 2: The main results are those obtained from the GCMS-analysis. It is not clear which compounds that are new, and the reliance of those identities, based on the present MS-resolution, would be low. The antioxidant part gives only one value for each of the two assays, and the antimicrobial part is also limited in results. I am not qualified to judge the contribution of the in silico part of the work.
Reply 2:
- The New compounds find in this study are added : Methyleugenol (15.41%), 1-Undecyne (3.4%), 2,6,10-Dodecatrien-1-ol, 3,7,11-trimethyl- (1.11%), Adamantane, 1,3-dimethyl- (0.92%), 2,5-Cyclohexadiene-1,4-dione, 2,6-bis(1,1-dimethylethyl)- (0.4%) and 5,9-Tetradecadiyne (0.32%).
- The antioxidant part and the antimicrobial part are given according to the find results.
- The goal of docking is to predict the predominant binding mode(s) of a ligand with a protein of known three-dimensional structure. Molecular docking (ligand-protein) is a key tool in structural molecular biology and computer-assisted antimicrobial, and antioxidant activities. Successful docking methods search high-dimensional spaces effectively and use a scoring function that correctly ranks candidate dockings. For a lot of reasons, in-silico research is essential to validation of compounds candidates against the studied activities.
Comment 3: The title seems to promise more than it keeps, and if published anywhere I would recommend to remove the word ‘new’.
Reply 3: title is revised.
Comment 4: Affiliation addresses are uncomplete.
Reply 4: completed.
Abstract:
Comment 5: Lines 29-31: This sentence is unclear. Do you mean: “The obtained results highlight the presence of 41 volatile compounds including two hitherto unreported compounds from this species: methyleugenol (15.41% of total volatiles), tricy-30 clo[2.2.1.0(2,6)]heptane, 1,3,3-trimethyl (10.68%).” Is this correct? If so, try to highlight what is the new contribution with this paper.
Reply 5: Revised.
Comment 6: What is meant by TAC? Write out the full name when this shortcut occurs for the first time in the manuscript. And what is meant by EAA?
Reply 6: all shortcuts are revised in the manuscript.
Comment 7: Use only significant numbers in all given results. It is odd to report a DPPH value at 1.247±0.056, IC50=19.05±1.87 and MIC at 16.67±5.77. I guess the uncertainty in these numbers is quite high?!!
Reply 7: checked and Revised.
Comment 8: It would be interesting to know in the abstract which microbe-strains were included in the tests.
Reply 8: microbe-strains used in the antimicrobial test included.
Comment 9: Lines 34-35: Unclear sentence: “The in-silico analysis revealed the presence of Compound Methyleugenol has two types of Pi-Alkyl…”
Reply 9: checked and Revised.
Comment 10: Lines 38-39: Remove this conclusion: “This study reveals some interesting results concerning the phytochemicals contained in GJEO and their effects on health.”
Reply 10: Removed.
Introduction:
Comment 11: This is not in accordance to what would be expected from a scientific article: A brief introduction to the actual area (plant background), an update to state-of-the-art (what is the present knowledge), gaps in knowledge, and a strategy of how to fill these gaps (motivation). In the present form it is a wordiness introduction. Simply too much verbiage.
Reply 11: thank you for your comment. The introduction is improved as requested.
Results and discussion:
Comment 12: The first paragraph contains no results! Remove!
Reply 12: the paragraph is removed.
Comment 13: Table 1 contain most of the data in this work, but based only on one analysis(!) The title claims ‘bioactive compound’, but there are no results that verifies that the single compounds are bioactive. The MS-data are of low-resolution, and doesn’t meet the criteria to claim new identities from this plant. There is no explanation of the ranges of SI and RSI, so the reader doesn’t necessarily know what a good match would look like. Beneath the table the is an explanation to Prob. Where is Prob?
Reply 13:
- The title has been revised.
- You’re right, the SI and RSI ranges are not necessary. We have removed it.
- The probability range is added.
Comment 14: All numbers with respect to antioxidant measures as well as those of the antimicrobial analyses operates with numbers with many decimal digits. This is not good. Use only significant numbers!
Reply 14: thank you. I considered your comment.
Comment 15: What is the news in 2.3?
Reply 15: this test was carried out to check the antimicrobial activities of G. jasminoides essential oil against various bacterial strains (gram -, gram +) and fungal strains. This leads to understand the biological effect of our samples and explore the data to forecast the pharmacokinetic properties, target engagement, and therapeutic potential of this botanical elixir, guiding future experimental endeavors and therapeutic applications.
Comment 16: Line 238: What is meant by a ‘good human protein’?
Reply 16: it was a typo.
Comment 17: References: Should be made in accordance to magazine’s instruction and guideline.
Reply 17: the references were revised according to journal guidelines.
Comment 18: Too much self-citation. Nine!!
Reply 18: Checked and revised.
Comment 19: Comments on the Quality of English Language.
Reply 19: English language was improved.
Comment 20: The introduction is characterized by wordiness and less content. In some cases in the Results and Discussion section it is hard to understand the meaning (e.i. line 156, 259-261).
Reply 20: You’re right. I have considered your remark.
Reviewer 3 Report
Comments and Suggestions for Authors
Antimicrobial Activities and Computational Assessment of Gardenia jasminoides Essential Oil
YOUR RECOMMENDATION: Minor Revision
This manuscript is an important addition to the literature by authors following the study of a bioactive compound derived from plants with a focus on the chemical characterization, antimicrobial activity, and molecular docking profile of Gardenia jasminoides essential oil (GJEO).
The study is well designed and fits within the field of the journal, especially in the context of natural antimicrobials as tools to limit the need for conventional antibiotics.
Major Comments:
In particular, the discussion should be more robust with respect to antimicrobial mechanisms, potential applications and contextualization of references;
The research is timely and relevant as there continues to be growing worldwide interest in the potential of plant-derived antimicrobials to treat multidrug-resistant (MDR) pathogens.
The work effectively combines chemical profiling, biological activity assessments, and in silico evaluations to offer a holistic perspective on the pharmaco-biological profiles of GJEO.
Systematic Analysis:
- GC-MS provides a robust chemical characterization.
- For antimicrobial testing, standardized in vitro protocols are followed, ensuring reproducibility of the results.
- The approach using molecular docking provides further mechanistic insight into the bioactivity.
Focus On Its Properties:
GJEO has antioxidant and antimicrobial properties, opening it up to numerous applications for pharmaceuticals, food preservation, and cosmetics formulations.
The section on mechanisms of action could be broadened to relate molecular docking findings on the lab bench to the broader world.
While the antimicrobial outcomes are promising, they need to be better related to the paradigm of MDR bacteria and biofilm formation.
Recent interest has focused on the potential of essential oils in combating antimicrobial resistance through targeting of bacterial communication systems (e.g., quorum sensing) and biofilm structure.
Suggested Integration:
“Essential oil from plants and related natural products have been researched more frequently for the alternative strategies against MDR bacteria owing to their antimicrobial activities.
Abrogating QSMs and biofilm formation is especially pertinent to the fight against chronic infection, and research activity in this area is focused on QSI as novel antimicrobial agents [28]. This is subtly in harmony with Hetta et al. (2024) towards quorum sensing inhibitors as alternative strategies against MDR bacteria.
Implications of Molecular Docking Results:
- The binding affinities of GJEO compounds must be compared to known antimicrobial agents to establish their potential therapeutic relevance.
- Can these insights lead to the rational design of novel antimicrobial formulations, e.g., nanoparticle-based delivery systems?
- Nanotechnology intelligent with natural antimicrobials characterized a solid global initiative, highlighting the plant-based molecules that could directly or indirectly combat resistant pathogens.
Expanding the Discussion on Biofilm Disruption Mechanism:
The discussion on how GJEO may affect biofilm formation can be elaborated further.
Plant-based approaches morphed to combat Acinetobacter baumannii, which is characterized by biofilm-forming pathogens that are more resistant to common antibiotics.
Refining Figures and Tables:
Mark all of the major bioactive compounds in the GC-MS chromatogram.
Molecular docking interactions could be improved with clearer figure labeling.
Language and Readability:
This manuscript is mostly well written, but could be improved with minor linguistic adjustments in some sections, e.g., for conciseness and clarity.
Technical details (e.g., molecular docking methodology) ought to be presented in a more simplified manner in order to increase the readability for non-specialist readers.
Minor revision
Author Response
We would like to express our gratitude to the reviewers’ for their constructive comments and valuable suggestions. Their feedback has greatly contributed to improving the quality of this manuscript. We have taken their recommendations into account and made the necessary revisions accordingly.
The revised parts are highlighted in yellow.
Antimicrobial Activities and Computational Assessment of Gardenia jasminoides Essential Oil
YOUR RECOMMENDATION: Minor Revision
This manuscript is an important addition to the literature by authors following the study of a bioactive compound derived from plants with a focus on the chemical characterization, antimicrobial activity, and molecular docking profile of Gardenia jasminoides essential oil (GJEO).
The study is well designed and fits within the field of the journal, especially in the context of natural antimicrobials as tools to limit the need for conventional antibiotics.
Major Comments:
Comment 1: In particular, the discussion should be more robust with respect to antimicrobial mechanisms, potential applications and contextualization of references;
The research is timely and relevant as there continues to be growing worldwide interest in the potential of plant-derived antimicrobials to treat multidrug-resistant (MDR) pathogens.
The work effectively combines chemical profiling, biological activity assessments, and in silico evaluations to offer a holistic perspective on the pharmaco-biological profiles of GJEO.
Reply 1: Thank you for your comment. I have revised this issue.
Comment 2:
Systematic Analysis:
- GC-MS provides a robust chemical characterization.
- For antimicrobial testing, standardized in vitro protocols are followed, ensuring reproducibility of the results.
- The approach using molecular docking provides further mechanistic insight into the bioactivity.
Reply 2: Thank you for your comment.
Comment 3:
Focus On Its Properties:
GJEO has antioxidant and antimicrobial properties, opening it up to numerous applications for pharmaceuticals, food preservation, and cosmetics formulations.
The section on mechanisms of action could be broadened to relate molecular docking findings on the lab bench to the broader world.
While the antimicrobial outcomes are promising, they need to be better related to the paradigm of MDR bacteria and biofilm formation.
Recent interest has focused on the potential of essential oils in combating antimicrobial resistance through targeting of bacterial communication systems (e.g., quorum sensing) and biofilm structure.
Suggested Integration:
“Essential oil from plants and related natural products have been researched more frequently for the alternative strategies against MDR bacteria owing to their antimicrobial activities.
Abrogating QSMs and biofilm formation is especially pertinent to the fight against chronic infection, and research activity in this area is focused on QSI as novel antimicrobial agents [28]. This is subtly in harmony with Hetta et al. (2024) towards quorum sensing inhibitors as alternative strategies against MDR bacteria.
Reply 3: I have considered your comments.
Implications of Molecular Docking Results:
- Comment 4:
- The binding affinities of GJEO compounds must be compared to known antimicrobial agents to establish their potential therapeutic relevance.
Reply 4: I have considered your comment.
- Comment 5:
- Can these insights lead to the rational design of novel antimicrobial formulations, e.g., nanoparticle-based delivery systems?
- Nanotechnology intelligent with natural antimicrobials characterized a solid global initiative, highlighting the plant-based molecules that could directly or indirectly combat resistant pathogens.
Reply 5:
- Yes, insights is possible to lead to the rational design of novel antimicrobial formulations, by using a nanoparticle-based delivery systems.
- You’re right, the conjugation of nanomaterials with antimicrobials may offer a solution to the global problem of resistance. This allows for specific and localized treatments which reduce the risk of adverse drug effects. These methods can help the recent developments in the application of nanotechnology to fight against antimicrobials. Recently, the development of nanoparticle-based vaccinations and novel antimicrobial medicines, as well as the use of artificial intelligence and machine learning–based tools to predict and detect bacteria and their resistance mechanisms.
Expanding the Discussion on Biofilm Disruption Mechanism:
Comment 6:
The discussion on how GJEO may affect biofilm formation can be elaborated further.
Plant-based approaches morphed to combat Acinetobacter baumannii, which is characterized by biofilm-forming pathogens that are more resistant to common antibiotics.
Reply 6: Thank you for your comments. I have added the mechanism of the GjEO effect on the biofilm formation.
Refining Figures and Tables:
Comment 7:
Mark all of the major bioactive compounds in the GC-MS chromatogram.
Reply 7: the major compounds are added in the chromatogram.
Comment 8:
Molecular docking interactions could be improved with clearer figure labeling.
Reply 8: I considered your comment, and I changed all 2D View in table 4 and table 5.
Language and Readability:
Comment 9:
This manuscript is mostly well written, but could be improved with minor linguistic adjustments in some sections, e.g., for conciseness and clarity.
Reply 9: the English language is improved.
Comment 10:
Technical details (e.g., molecular docking methodology) ought to be presented in a more simplified manner in order to increase the readability for non-specialist readers.
Reply 10: we have revised and clarified this section.
Reviewer 4 Report
Comments and Suggestions for Authors
Standard deviation not to be cited in texts (only in figures or tables)
Song et al., (2013) is not the correct form. Review all references
Chlorinated compounds are not acceptable propositions. Discard these propositions.
You used GC-MS for percentages with derivatization. Since this is not the usual method to quantify essential oils (GC-FID without derivatization), you can compare between your samples but it is better to say volatile extract instead of essential oil.
There are protocols that are not detailed enough (example antioxidant). Review the protocols and cite a bibliographic reference for each protocol.
Table 1. Bioactive compound identified in GJEO through GC-MS. Change the title because not all compounds are bioactive (example alkanes). Put identified compounds. The identification is not reliable because you have chlorinated compounds, nitrogen compounds, alkanes while they are terpenes. In addition, you did not use the retention index (validation of terpenes) and the GC-FID for reliable quantities. The percentage of similarity is not sufficient. Review all the identification and quantification.
Table 3. add the standard deviations of the antibiotic values. Add the concentrations of the samples and antibiotics tests. How did you judge that it is resistant? (Rs resistant)
All tables: same number of digits after the decimal point
Author Response
We would like to express our gratitude to the reviewers’ for their constructive comments and valuable suggestions. Their feedback has greatly contributed to improving the quality of this manuscript. We have taken their recommendations into account and made the necessary revisions accordingly.
The revised parts are highlighted in yellow.
Comment 1: Standard deviation not to be cited in texts (only in figures or tables)
Reply 1: Revised.
Comment 2: Song et al., (2013) is not the correct form. Review all references
Reply: All references are revised.
Comment 3: Chlorinated compounds are not acceptable propositions. Discard these propositions.
Reply 3: Chlorinated compounds are not a bioactive compound in the context of medicinal plant chemistry or essential oils. Chlorinated compounds falls into the category of volatile organic compounds (VOCs) due to its low molecular weight and non-polar nature, which makes it susceptible to evaporation at room temperature. However, the compound list was revised.
Comment 4: You used GC-MS for percentages with derivatization. Since this is not the usual method to quantify essential oils (GC-FID without derivatization), you can compare between your samples but it is better to say volatile extract instead of essential oil.
Reply 4: Thank you for your comment. I considered your suggestion.
Comment 5: There are protocols that are not detailed enough (example antioxidant). Review the protocols and cite a bibliographic reference for each protocol.
Reply 5: thank you for your comment. The protocols are described briefly because they are known and previously cited in our previous works.
Comment 6: Table 1. Bioactive compound identified in GJEO through GC-MS. Change the title because not all compounds are bioactive (example alkanes). Put identified compounds. The identification is not reliable because you have chlorinated compounds, nitrogen compounds, alkanes while they are terpenes. In addition, you did not use the retention index (validation of terpenes) and the GC-FID for reliable quantities. The percentage of similarity is not sufficient. Review all the identification and quantification.
Reply 6: the title is changed as suggested. And the data is revised and checked.
Comment 7: Table 3. add the standard deviations of the antibiotic values. Add the concentrations of the samples and antibiotics tests. How did you judge that it is resistant? (Rs resistant)
Reply 7:
- The standard deviations of the antibiotic values were added.
- The concentrations of the samples and antibiotics tests were added.
- To judge that a microbial strain is resistant was done according to Comité de l'Antibiogramme de la Société Française de Microbiologie (CASFM) and EUCAST or CLSI.
Ampicillin (10 µg/disc):
Microorganism |
Critical Diameter (mm) |
Critical MIC (µg/ml) |
E. coli |
S ≥ 17 ; R ≤ 13 |
S ≤ 8 ; R ≥ 32 |
P. aeruginosa |
Naturally resistant |
Not applicable |
S. aureus |
S ≥ 29 ; R ≤ 28 |
S ≤ 0.25 ; R ≥ 0.5 |
Penicillin (10 µg/disc):
Microorganism |
Critical Diameter (mm) |
Critical MIC (µg/ml) |
E. coli |
Naturally resistant |
Not applicable |
P. aeruginosa |
Naturally resistant |
Not applicable |
S. aureus |
S ≥ 29 ; R ≤ 28 |
S ≤ 0.12 ; R ≥ 0.25 |
Comment 8: All tables: same number of digits after the decimal point.
Reply 8: revised.
Round 2
Reviewer 1 Report
Comments and Suggestions for Authors
Manuscript has been improved following revision.
Author Response
Thank you for your kind comment.
Reviewer 2 Report
Comments and Suggestions for Authors
Though the quality of the manuscript has been improved in accordance to my previous comments, there is no new contribution that lifts its originality, significance or overall interest to the readers; there are no new experiments included! Thus, I end up with an overall low merit of the manuscript and cannot advise it be be published in this journal.
Author Response
Comment 1: Though the quality of the manuscript has been improved in accordance to my previous comments.
Reply 1: Thank you for your comment.
Comment 2: There is no new contribution that lifts its originality, significance or overall interest to the readers; there are no new experiments included! Thus, I end up with an overall low merit of the manuscript and cannot advise it to be published in this journal.
Reply 2:
This research advances the scientific understanding of Gardenia jasminoides essential oil by analyzing its chemical profile, assessing its antioxidant and antimicrobial properties, and performing in silico evaluation using the most known methods and techniques. The obtained results showed:
- The presence of new compounds.
- Biological activity revealed remarkable results.
- The results of the in silico evaluation gave interesting predictions that can be used in the pharmaceutical field.
Reviewer 4 Report
Comments and Suggestions for Authors
The list of 41 identified compounds must be reduced by citing the number without the alkanes that may come from the injection solvent.
Table 2; review the μL/mL unit that does not match with BHT and ascorbic acid. Moreover, a doubt about the value of ascorbic acid.
Table 3 define how you judged the resistance of the strains to the essential oil. Rs: resistant
In the conclusion, remove "new" because nothing indicates that these are new compounds.
Author Response
Comment 1: The list of 41 identified compounds must be reduced by citing the number without the alkanes that may come from the injection solvent.
Reply 1: This issue has already been revised within the last revision. I have checked it again. The alkane compounds have been removed: Pentacosane, Hexacosane, Octacosane and Adamantane,1,3-dimethyl. A total of 25 volatile compounds are identified.
Comment 2: Table 2; review the μL/mL unit that does not match with BHT and ascorbic acid. Moreover, a doubt about the value of ascorbic acid.
Reply 2:
Thank you for your remarks. The comment is relevant. It's a typo.
- The unit is revised.
- The value of ascorbic acid must be in (748.10±20.78 µg/100mL) and it is revised. Lines 202-207.
Comment 3: Table 3 define how you judged the resistance of the strains to the essential oil. Rs: resistant.
Reply 3: Strain resistance to essential oils is considered to exist when the zone of inhibition is less than 9 mm.
Comment 4: In the conclusion, remove "new" because nothing indicates that these are new compounds.
Reply 4: The obtained results highlight the presence of 25 volatile compounds including 5 new detected compounds: Methyleugenol (15.41%), 1-Undecyne (3.4%), 2,6,10-Dodecatrien-1-ol, 3,7,11-trimethyl- (1.11%), Adamantane, 1,3-dimethyl- (0.92%), 2,5-Cyclohexadiene-1,4-dione, 2,6-bis(1,1-dimethylethyl)- (0.4%) and 5,9-Tetradecadiyne (0.32%).